# Fine Mapping and Functional Analysis of Major Regulatory Genes of Soluble Solids Content in Wax Gourd (*Benincasa hispida*)

**DOI:** 10.3390/ijms23136999

**Published:** 2022-06-23

**Authors:** Wenting Wu, Peng Wang, Xiaochun Huang, Liwen Su, Haixuan Lv, Jiquan Gou, Zhikui Cheng, Lianlian Ma, Wenjin Yu, Zhengguo Liu

**Affiliations:** 1College of Agriculture, Guangxi University, Nanning 530004, China; 2017391034@st.gxu.edu.cn (W.W.); 2017302003@st.gxu.edu.cn (X.H.); 2017302009@st.gxu.edu.cn (L.S.); 2017391023@st.gxu.edu.cn (H.L.); 1917391004@st.gxu.edu.cn (J.G.); 1910302002@st.gxu.edu.cn (Z.C.); 1917391016@st.gxu.edu.cn (L.M.); yuwjin@gxu.edu.cn (W.Y.); 2Institute of Vegetable Research, Guangxi Academy of Agricultural Sciences, Nanning 530004, China; wangpeng@gxaas.net

**Keywords:** wax gourd, soluble solids content, BSA, QTL, NADP-dependent malic enzyme

## Abstract

Soluble solids content (SSC) is an important quality trait of wax gourd, but reports about its regulatory genes are scarce. In this study, the SSC regulatory gene *BhSSC2.1* in wax gourd was mined via quantitative trait locus (QTL) mapping based on high-density genetic mapping containing 12 linkage groups (LG) and bulked segregant analysis (BSA)-seq. QTL mapping and BSA-seq revealed for the first time that the SSC QTL (107.658–108.176 cM) of wax gourd was on Chr2 (LG2). The interpretable phenotypic variation rate and maximum LOD were 16.033% and 6.454, respectively. The QTL interval contained 13 genes. Real-time fluorescence quantitative expression analysis, functional annotation, and sequence analysis suggested that *Bch02G016960*, named *BhSSC2.1*, was a candidate regulatory gene of the SSC in wax gourd. Functional annotation of this gene showed that it codes for a NADP-dependent malic enzyme. According to *BhSSC2.1* sequence variation, an InDel marker was developed for molecular marker-assisted breeding of wax gourd. This study will lay the foundation for future studies regarding breeding and understanding genetic mechanisms of wax gourd.

## 1. Introduction

Soluble solids content (SSC) is an important quality trait of several melon crops, and Brix is commonly used as an indicator. It is an important factor affecting fruit quality. It is also a complex quantitative trait [1] controlled by multiple genes. It mainly constitutes soluble sugars, small amounts of organic acids, soluble proteins, and minerals.

Studies of the SSC of wax gourd (*Benincasa hispida* (Thunb.) Cogn., 2n = 2x = 24), an annual trailing herb that belongs to *Cucurbitaceae* originating from China and East India and widely distributed in tropical, subtropical, and temperate regions of Asia [2], mainly focus on cultivation measures and physiological indices; however, related molecular genetic research is rarely performed. Wax gourd varieties are a main staple vegetable in summer and autumn in China, are rich in nutrients, and have both medicinal and health benefits [3]. Wax gourd has good storage resistance and long shelf life, so it is one of the main vegetables that regulate the annual equilibrium supply in the market [4]. During the fruit development of chieh-qua (*B. hispida* var. chieh-qua How.), the soluble sugar content of fresh samples increases gradually as the fruit develops. The growth rate increases slowly from 10 days after pollination (DAP) to 30 DAP, and increases significantly from 30 DAP to 40 DAP. Sugar mainly accumulates in the later growth stages [5]. MX2-ADI-AD is the best genetic model for SSC; the main gene heritability of SSC is high [1].

Key regulatory genes must be identified in high-quality molecular design breeding of related crops, and their regulatory mechanisms must be clarified. The quality of wax gourd fruits, which is affected by SSC, is important for consumers’ acceptance and preference [6]. Other studies of SSC of *Cucurbitaceae* crops applied forward genetics, homologous cloning, and other techniques. For example, in melon, more than 60 QTL loci related to SSC are reported, but most of them have a low contribution to phenotype [7,8,9,10,11,12]. *SUCQSC5.1* is the only fine-mapped QTL for sugar accumulation in melon, which is located on Chr5 and can reduce SSC by 18%, and sucrose content by 34%. The candidate gene *MELO3C014519* encodes a putative bel1-like homeodomain protein, which may affect sugar accumulation by influencing growth processes [13]. The tonoplast monosaccharide transporter family can regulate sugar through vacuoles [14]. In the cloning and functional analysis of melon tonoplast sugar transporter, CmTST2 plays an important role in sugar accumulation. Overexpression of CmTST2 may delay maturation through signal feedback regulation, resulting in the accumulation of more sugar [15]. Similarly, the tonoplast sugar transporters ClTST2 and CsTST2 in watermelon and cucumber are associated with sugar accumulation in these fruits [16,17].

Hashizume et al. [18] used RAPD, RFLP, ISSR, and isozyme markers in watermelon to construct linkage maps and identify QTLs for SSC for the first time; they found QTL on linkage group (LG) 8, accounting for 18.8% of phenotypic variation. Since then, researchers extensively studied the SSC of watermelon and identified more than 30 related QTLs through genetic linkage maps [19,20,21]. In addition, the SSC regulation mechanism in watermelon was investigated using transcriptomics and reverse genetics. Using the watermelon inbred lines 203Z and SW, Umer et al. [22] found that fructose is the main sugar in watermelon, and *WMFBP-2*, involved in fructose breakdown, is significantly and positively correlated with fructose. They then used transcriptome profiles to identify seven genes involved in sugar and organic acids metabolism; specifically, *Cl97C01G000640*, *Cl97C05G087120*, and *Cl97C01G018840* are identified as sugar transporters, and *Cl97C03G064990* is identified as a sucrose synthase [23]. The NAC transcription factor *ClNAC68* is a transcriptional repressor that binds to the promoters of *ClINV* and *ClGH3.6* in watermelon to repress their expression and positively regulate sugar accumulation [24].

Bulked segregant analysis (BSA) and QTL mapping with high-density genetic maps are effective methods to locate the main genetic regions of target traits, and are widely used in horticultural crops, such as watermelon [25], melon [26], peach [27], cabbage [28], and cucumber [29]. In wax gourd, the fine mapping of peel color and fruit shape candidate genes was completed via BSA [30,31]. The high-density genetic map of wax gourd constructed using SLAF-seq identified QTLs related to peel color, fruit weight, fruit length, fruit diameter, and flesh thickness [32,33]. However, the genetic localization of SSC-related candidate genes in wax gourd has not yet been reported. Therefore, in this study, we used the high-density genetic map constructed with SL-7 and XDJQ-1 and phenotypic data of an F_2_ population of hybrids to map SSC-related QTLs in wax gourd. Individual plants showing extreme traits were selected from an F_2_ population for BSA verification of QTL mapping results. Based on QTL mapping and BSA results, the candidate genes regulating SSC in wax gourd were excavated, thereby providing a theoretical basis for breeding and understanding genetic mechanisms of wax gourd in the future.

## 2. Results

### 2.1. Soluble Sugar Content and SSC of SL-7 and XDJQ-1

To observe the changes in soluble sugar content and composition during wax gourd development, we measured the contents of glucose, fructose, and sucrose 5, 10, 15, 20, 25, and 30 DAP of SL-7 and XDJQ-1 and then calculated the total sugar content. We found that the soluble sugars of SL-7 and XDJQ-1 were mainly glucose and fructose; however, the sucrose content was almost undetectable, and therefore not considered. At different fruit development stages, the glucose and fructose contents in SL-7 were significantly higher than those in XDJQ-1. The glucose content in SL-7 increased continuously from days 5 to 20, reached the maximum (20.78 mg/g) on day 20, and gradually decreased until day 30. The glucose content in XDJQ-1 showed a slightly different trend from that of SL-7; that is, the highest value (9.78 mg/g) was obtained on day 15, but it decreased on day 30 (Figure 1a). Changes in fructose contents of SL-7 and XDJQ-1 were the same. The fructose content increased from days 5 to 15, reached the highest value (22.37 mg/g for SL-7 and 11.44 mg/g for XDJQ-1) on day 15, and gradually decreased until day 30 (Figure 1b). During wax gourd development, the total sugar content initially increased and subsequently decreased. SL-7 and XDJQ-1 reached maximum values of 43.14 and 21.22 mg/g on days 20 and 15, respectively. The maximum difference between them was observed 20 DAP (Figure 1c).

In actual production, the harvest time of commercial wax gourd is generally 20 DAP. In our study, the sugar content difference was the highest 20 DAP; therefore, 20 DAP was selected to measure the SSC of wax gourd. The results show that the average SSCs of SL-7 and XDJQ-1 were 5.3% Brix and 2.2% Brix, respectively. The SSC significantly differed between the parents; the average SSC of F1 plants was 3.3% Brix (Figure 1d). The SSC of 1493 F_2_ plants was measured 20 DAP and subsequently analyzed using SPSS 25.0 to draw a frequency distribution histogram (Figure 1e). The SSC of F_2_ plants was generally between that of their parents, showing a normal distribution trend and consistency with quantitative genetic characteristics.

### 2.2. BhSSC2.1 Mapping and Candidate Gene Screening

A high-density genetic map of wax gourd containing 12 LGs with a total map distance of 1243.87 cM and an average map distance of 0.33 cm was constructed to identify candidate SSC gene regions of wax gourd. The linkage map of Chr2 is shown in Figure 2a, which contains 302 Bins in total, with a total genetic distance of 121.81 cM and an average map distance of 0.41 cM. Among them, the proportion of Gap < 5 cM is 99%. The QTL for the SSC of wax gourd was identified on Chr2 with a genetic distance of 107.658–108.176 cM (0.518 cM) between Block6199 and Block6269 using the high-density genetic map and phenotypic data of the F_2_ population (Figure 2b). The candidate range was 58,157,714–58,760,208 (0.60 Mb). The maximum LOD of this QTL and the phenotypic variation rate were 6.454 and 16.033%, respectively.

The candidate genes in this QTL interval were functionally annotated on the basis of the unpublished genome-wide annotation information of wax gourd genome GX-19. The results showed 13 candidate genes in this QTL interval and 12 annotated genes in the database (Figure 2c). Of the 13 genes, two (*Bch02G016940* and *Bch02G016960*) had nonsynonymous mutations in their coding regions (Table 1). *Bch02G016940* was annotated as putative ABC transporter B family member 8, which is related to ATP catabolism and transmembrane transport. *Bch02G016960* was annotated as NADP-dependent malic enzyme (NADP-ME), which is a key enzyme in malic acid metabolism and closely related to the tricarboxylic acid cycle (TCA) pathway. The functional annotation and sequence alignment results preliminarily indicated that *Bch02G016960*, named *BhSSC2.1*, might be the key gene regulating the SSC of wax gourd.

### 2.3. BSA-Seq Verification of BhSSC2.1 Localisation

Thirty plants each with a high and low SSC were selected from the F_2_ population obtained from the hybridization of SL-7 and XDJQ-1 to construct a mixed pool of extreme traits for BSA-seq. Euclidean distance (ED) algorithm was used to analyze the correlation between mixed pools and obtain the BSA location results. Before association analysis, SNPs were filtered to obtain 1,320,495 trusted high-quality SNPs. According to the depth of each SNP in the two mixed pools, the ED value of each locus was calculated and fitted, and the fitted value was 0.32. According to the association threshold, a gene candidate region was found on Chr2, ranging from 49,990,000 bp to 64,930,000 bp, having a length of 14.94 Mb, and containing 433 genes (Figure 3). The comparison of the *BhSSC2.1* mapping results with the candidate regions of BSA revealed that *BhSSC2.1* was close to the peak of the BSA mapping interval. The two results were consistent with each other, indicating that the *BhSSC2.1* mapping results were reliable.

### 2.4. Real-Time Fluorescence Quantitative Expression Analysis

The relative differences in the expression of the 13 candidate genes in the QTL range were analyzed in the flesh of SL-7 and XDJQ-1 fruits at 20 DAP. The results show that only the expression level of *Bch02G016960* was significantly different in the flesh of the parents at 20 DAP (Figure 4), indicating that *Bch02G016960* could be a key gene regulating the SSC of wax gourd. This result also verified the localization results of *BhSSC2.1*.

### 2.5. Gene Sequence Analysis

The CDS sequence of *Bch02G016960* was searched according to the GX-19 reference genome to further verify whether *Bch02G016960* is a candidate gene for the SSC of wax gourd, and the full-length CDS sequence in SL-7 and XDJQ-1 was amplified (Appendix A). The sequencing results of the parents were compared using DNAMAN software. As shown in Figure 2d, the CDS length of *Bch02G016960* was 1776 bp, the sequence of SL-7 was consistent with that of GX-19, and the CDS sequence of *Bch02G016960* of XDJQ-1 had six base mutations. One base at 1172 bp underwent nonsynonymous mutation, replacing cytosine (C) with adenine (A), and the corresponding amino acid was transformed from threonine (T) to lysine (K; Appendix A).

### 2.6. Development of InDel Markers for Molecular Marker-Assisted Breeding

An InDel marker was developed in the intron based on differences in the *Bch02G016960* sequences in the parents, and the primer was designed using Primer Premier 5. Fifty wax gourd inbred lines with extreme SSC differences were selected to verify InDel labeling: 20 with high SSC (SSC > 4.5%) and 30 with low SSC (SSC < 2.5%; Appendix A). The results show that 20 high-SSC bands were consistent with SL-7 bands, while 21 low-SSC bands were consistent with XDJQ-1 bands, and 9 were inconsistent with XDJQ-1 bands (Figure 5). The coincidence rate between genotype and phenotype was 82%; the InDel marker can be used in the SSC molecular marker-assisted breeding of wax gourd.

## 3. Discussion

Although previous studies demonstrated that SSC, an important quality trait of wax gourd, affects the taste and consumer preference, they were mostly limited to physiology and cultivation; only a few studies investigated the molecular mechanism in wax gourd [5,6]. In this study, a pair of parents with significant differences in SSC was used to map the main regulatory genes of SSC in wax gourd via QTL and BSA-seq methods. The SSC in fruits and vegetables is mostly composed of soluble sugars, mainly glucose, fructose, and sucrose [34]. We found that soluble sugars that accumulated at 5–30 DAP were mainly glucose and fructose, and they changed dynamically during development; particularly, they initially increased and subsequently decreased. Among *Cucurbitaceae* crops, the changes in total sugar content in watermelon [35] and melon [36] during fruit development differed greatly; in some varieties it increased first and then decreased, while in others it increased continually. In wax gourd, Huang and Xue [37] observed a trend of sugar content rising first and then decreasing, which is consistent with our research results. In contrast, Liu et al. [5] observed that the total sugar content of winter melon continued to increase; this difference may be related to the different materials we selected and the different ways sugar accumulates.

In our study, the high-density genetic map and QTL mapping analysis revealed that the candidate regions of the major regulatory genes for SSC in wax gourd were at 107.658–108.176 cM of Chr2, and that the QTL interval contained 13 genes. The sequence analysis, functional annotation, and fluorescence quantitative expression analysis of the 13 genes in the QTL interval suggested that *Bch02G016960*, named *BhSSC2.1*, might be the main regulatory gene of SSC. The gene functional annotation revealed that *BhSSC2.1* functions as a NADP-ME. Subsequently, we performed BSA-seq to verify the localization results of *BhSSC2.1*. We then performed BSA-seq to verify the localization results of *BhSSC2.1*. Numerous QTLs related to SSC and sugar content were identified in watermelon and melon using the F_2_ population [38], F_2:3_ population [21], and recombinant inbred lines (RILs) [12] of cucurbit plants. The only QTL with fine mapping is *SUCQSC5.1* in melon [10]. In this study, a high-density genetic map was used to identify QTL loci for SSC in wax gourd for the first time, and fine mapping was completed.

SSC is a general term for all water-soluble compounds, including soluble sugars, organic acids, soluble proteins, and minerals. Changes in these components affect the SSC, among which, the soluble sugar content is the most important. Changes in sugar metabolism during fruit development can affect changes in soluble sugars and the proportion of various sugars [39]. The sucrose invertase genes *Sucr* [40] and *Lin5* [41] influence sugar accumulation by affecting invertase activity in tomato. Several enzymes involved in sugar metabolism, such as Beta-glucosidase and bidirectional sugar transporter SWEET10-like, play the positive role of watermelon rootstock in the sweetness of bottle gourd fruits [42]. In addition to sugar metabolism-related genes that regulate sugar accumulation, sugar accumulation-related genes have been found. Sugar accumulation is negatively affected by vacuolar processing enzyme (VPE) [43] and inhibited by the loss of the synthesis of the plant pigment bilitin (P φ B) in tomato fruits [44]. Putative bel1-like homeodomain protein may affect sugar accumulation by influencing the growth processes in melon [12]. These results suggest that sugar accumulation involves complex regulatory networks and is not regulated by simple glucose metabolism.

NADP-ME is a key enzyme involved in malic acid metabolism in fruits [45] and is associated with malic acid degradation during fruit ripening [46]. Malic acid is an intermediate product of the TCA cycle, which is closely related to its synthesis and degradation [47]. The TCA cycle is the ultimate pathway of glucose metabolism, which begins with the condensation of oxaloacetic acid (OAA) and acetyl CoA, proceeds through a series of oxidation steps, releases two carbon atoms as CO_2_, and ends with OAA regeneration [48]. In plants, the acetyl CoA required for this cycle is usually obtained from glycolytic products through the action of pyruvate dehydrogenase on mitochondrial pyruvate [48]. Pyruvate can be transfused from the cytoplasm or synthesized by the action of NADP-ME on malic acid. In the cytoplasm, hexose generates phosphoenolpyruvate (PEP) through glycolysis. Under the action of PEP carboxylase and malate dehydrogenase, PEP generates malic acid through a series of reactions, and finally degrades to pyruvate under the action of NADP-ME [49]. In summary, NADP-ME is closely related to glucose metabolism and malic acid metabolism in fruits. Malic acid content has a significant negative correlation with sugar content in apple [47], passion fruit [50], and watermelon [51]. This result indicates that malic acid and sugar compete for carbon sources [47]. The combined action of NADP-ME and phosphoenolpyruvate carboxykinase promotes a rapid decrease in malic acid content and a rapid increase in SSC at maturity [52]. Therefore, NADP-ME can affect SSC accumulation in fruits in various ways. In our study, the change in NADP-ME in low-SSC wax gourd might result in a decrease in NADP-ME activity, further decreasing the malic acid degradation rate and reducing the sugar synthesis rate, thus affecting SSC accumulation. However, the synthesis and decomposition of sugars and acids are complex processes, and interactions between sugars and acids remain unclear; hence, specific molecular mechanisms are yet to be further verified.

SSC in wax gourd must be improved through quality breeding. In fruits and vegetables, SSC is often used to indicate sugar content because it is mainly composed of soluble sugar and can be quickly measured using a refractometer [53]. The higher the total sugar content, the better the taste and nutrition of wax gourd [6]. Therefore, breeding wax gourd with high SSC and sugar content is important. In production, improving fruit SSC is a complex and lengthy process, and the development of molecular markers that can be used for production can greatly contribute to production and breeding. In this study, we developed an InDel marker and identified 50 wax gourd inbred lines with significantly different SSC. The genotype and phenotype coincidence rate was 82%, which could be used in molecular marker-assisted breeding. Future studies should be conducted to fine-locate and clone other major genes for SSC in wax gourd, develop submarkers closely linked to each major gene, and comprehensively apply molecular markers associated with each major gene to enhance SSC in wax gourd for molecular marker-assisted breeding.

## 4. Materials and Methods

### 4.1. Experimental Materials

In this study, wax gourd SL-7 with a high SSC (5.3% Brix) and XDJQ-1 with a low SSC (2.2% Brix) were respectively selected as the male and female parents to construct an F_2_ segregating population. In spring 2020, 273 individual plants from the F_2_ population were planted. In autumn 2020, 1493 individual plants from the F_2_ population were planted. All plants were transplanted in the ground with a row spacing of 0.5 m × 1.2 m after they grew two true leaves in hole seedling trays. Conventional cultivation and management were then conducted. They were planted at the experimental base in Shajing, Nanning, Guangxi, China (108°51′ E and 22°48′ N).

### 4.2. Measurement of SSC in Wax Gourd Flesh

The date of pollination was recorded for each fruit. The fruits of two parents, F_1_ and F_2_, were harvested 20 DAP and pressed into homogenates using a high-speed homogenizer. The SSC was measured using a handheld refractometer. Two of the parents and F_1_ were measured, using 15 plants respectively, each measuring 3 melons and each melon was measured 3 times. F_2_ plants were measured using 3 melons per plant with 3 measurements per melon.

### 4.3. Determination of Glucose, Fructose, and Sucrose

The two parents were picked and pressed into homogenates using a high-speed homogenizer 5, 10, 15, 20, 25, and 30 days after pollination. Next, 5 g was weighed and put into a 50 mL tube; ultra-pure water was added to 50 mL, and this was placed in a water bath at 80 °C for 30 min, shaking every 10 min. The mixture was cooled to room temperature and filtered. The filtrate (5 mL) was passed through a solid phase extraction small column with a 0.22 pore size filter membrane. The glucose, fructose, and sucrose contents of the filter were determined by ion chromatography [22].

### 4.4. DNA Extraction

Three weeks after planting, the fresh young leaves of the F_1_ and F_2_ parents were collected in a 2 mL centrifuge tube and stored at −80 °C for DNA extraction. DNA was extracted using the improved CTAB method [54]. The concentration and purity of the DNA were determined using an ultra-micro spectrophotometer, and the quality of DNA was examined through 1.2% agarose gel electrophoresis.

### 4.5. QTL Analysis

QTL mapping was performed based on the high-density genetic map of wax gourd constructed by the research group. The genetic map was based on 195 F_2_ plants (SL-7 × XDJQ-1) as the genetic population for mapping, containing 323107 SNP markers, divided into 3720 Bins, a total of 12 LG, with a total length of 1243.87 cM and an average distance of 0.33 cM. Using R/QTL software combined with F_2_ population phenotypic data, the SSC was analyzed using a composite interval mapping method. First, the threshold of the LOD value was determined by PT test 1000 times, and the LOD thresholds corresponding to confidence levels of 0.99, 0.95, and 0.90 were considered in turn; if there was no positioning interval, the PT test result was not considered. The threshold was manually lowered to 3.0, if not, 2.5 was considered. After determining the threshold, it was determined that there are QTL sites related to SSC in this range.

### 4.6. BSA Location

From a total of 195 resequenced F_2_ single plants, 30 plants with high SSC (>4.8% Brix) and 30 plants with low SSC (<2.4% Brix) were selected. Resequencing data were then combined to construct a mixed pool of resequencing data for extreme traits. Association analysis was conducted between a high–low mixed pool of SSC and two parent pools with GX-19 as the reference genome (unpublished). SNP and InDel were detected using GATK software, and correlation between mixed pools was analyzed using the ED algorithm.

### 4.7. Candidate Gene Predictive Analysis

Multiple databases (NR (https://www.ncbi.nlm.nih.gov/refseq/about/nonredundantproteins/ accessed on 25 May 2022), Swiss-PROt (https://ngdc.cncb.ac.cn/databasecommons/database/id/5614 accessed on 25 May 2022), GO (http://geneontology.org/ accessed on 25 May 2022), KEGG (https://www.genome.jp/kegg/ accessed on 25 May 2022), and COG (https://www.ncbi.nlm.nih.gov/COG/ accessed on 25 May 2022)) were used to annotate the coding genes in candidate regions obtained from BSA and QTL analysis using BLAST to screen the candidate genes likely involved in the regulation of SSC traits.

### 4.8. RNA Extraction and Candidate Gene Cloning and Sequencing

According to the CDS of the candidate genes, Primer Premier 5.0 was used to complete the design of primers for gene cloning (Appendix A). Total RNA was extracted from SL-7 and XDJQ-1 leaves using the Eastep^®^ Super Total RNA Extraction Kit (Promega, Beijing, China) according to the manufacturer’s instructions. The target gene was amplified using 2 × Phanta^®^ Max Master Mix (Vazyme, Nanjing, China). Agarose gels with target bands were recovered and purified using Axyprep DNA gel extraction kits (Axygen, Union City, CA, USA), and qualified gel recovery products were tested for further experiments. The zero-background Ptopo-Blunt cloning kit was used (CV16); the recombinant plasmid was constructed using Aidlab, and transformed into *Escherichia coli* DH5α competent cells. Positive monoclones were selected and sequenced. DNA and amino acid sequences were compared using DNAMAN V.9.

### 4.9. RT-qPCR Analysis of Candidate Genes

Real-time fluorescence quantitative PCR (RT-qPCR) was used to quantify the specific expression of candidate genes in parental flesh. Pulp RNA was extracted from parent 20 DAP and reversely transcribed into cDNA. Quantitative fluorescence analysis was performed using the Applied Biosystems7500 real-time PCR instrument (Foster City, CA, USA). Actin (*Bch10G018990*) and candidate genes were designed using Primer Premier 5.0 (Appendix A). Three biological replicates were performed for each gene, and the relative gene expression was evaluated via the 2^−ΔΔCt^ method [31].

### 4.10. Molecular Marker Development

According to the difference in candidate gene sequences, a pair of InDel markers that can be used in molecular-assisted breeding were designed; primer sequences are shown in Appendix A. See the attached table for primer sequences. Parents SL-7, XDJQ-1, F_1_, and 50 wax gourd inbred line materials were used to verify the markers, including 20 high SSC materials and 30 low SSC materials.

## Figures and Tables

**Figure 1 ijms-23-06999-f001:**
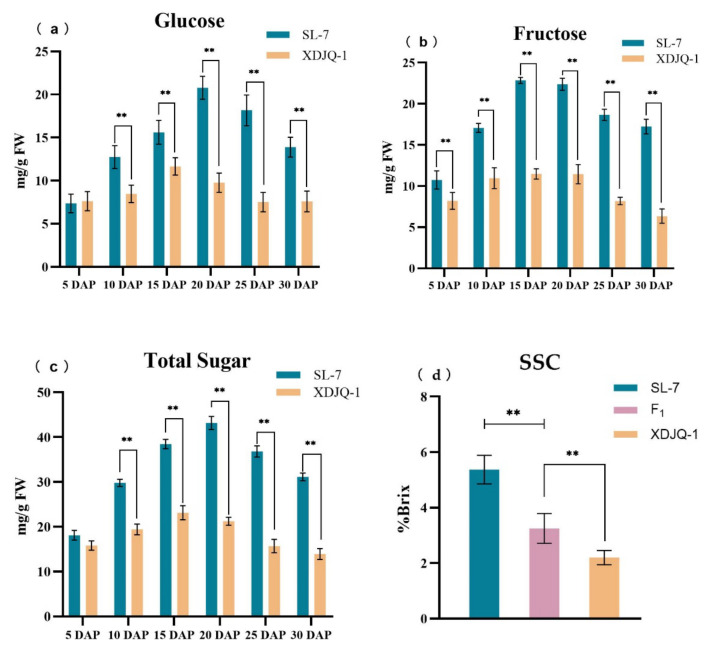
SSC and sugar content of wax gourd. (**a**) Glucose content of SL-7 and XDJQ-1 at 5–30 DAP. (**b**) Fructose content of SL-7 and XDJQ-1 at 5–30 DAP. (**c**) Total sugar content of SL-7 and XDJQ-1 at 5–30 DAP. (**d**) SL-7, XDJQ-1, and F_1_ at 20 DAP. ** *p* < 0.01 (**a**–**d**). (**e**) SSC frequency distribution histogram of 1493 F_2_ strains at 20 DAP.

**Figure 2 ijms-23-06999-f002:**
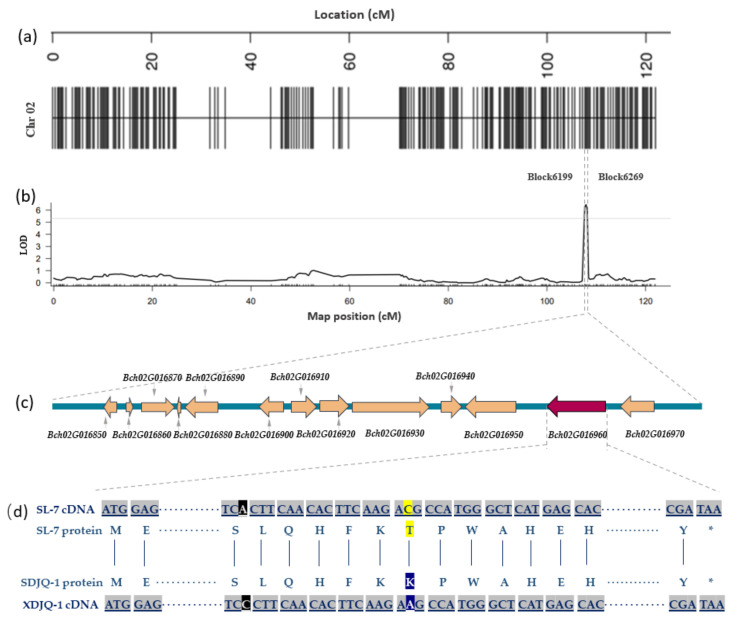
QTL analysis. (**a**) A high-density genetic map of Chr02. (**b**) QTL analysis of SSC on Chr02 (LG2). (**c**) 13 candidate genes in QTL interval. (**d**) Comparison of coding sequence (CDS) and protein sequence between the parents of *Bch02G016960*. ***** Translation termination.

**Figure 3 ijms-23-06999-f003:**
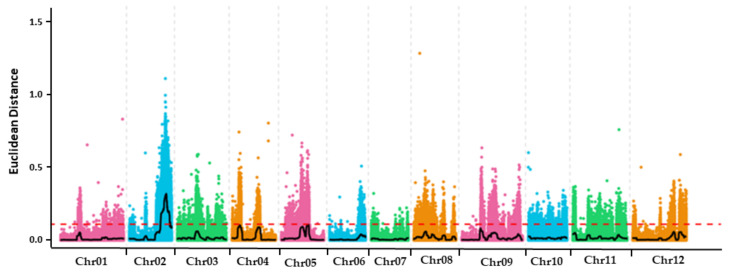
BSA location results of the SSC of wax gourd based on Euclidean distance (ED) association algorithm. The colored dots represent the ED value of each SNP, the black line represents the fitted ED value, and the red dotted line represents the significance association threshold.

**Figure 4 ijms-23-06999-f004:**
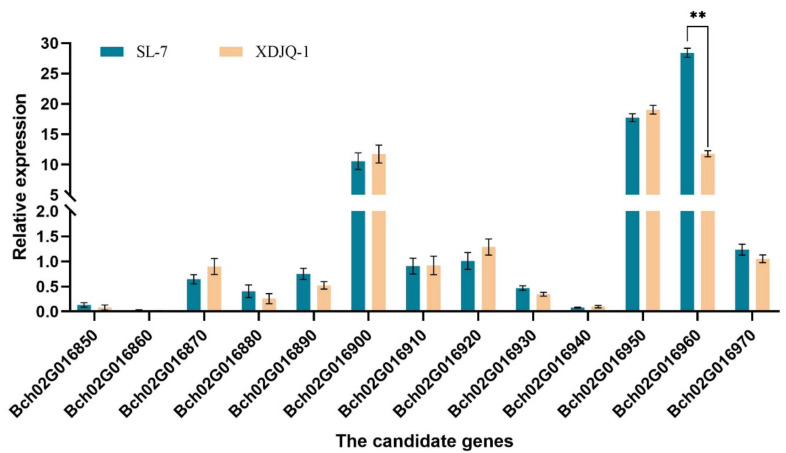
Expression levels of 13 candidate genes in the pulp of SL-7 and XDJQ-1 at 20 DAP. ** *p* < 0.01.

**Figure 5 ijms-23-06999-f005:**
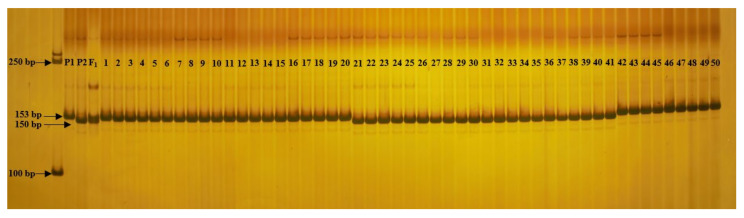
InDel is marked for verifying 50 parts of wax gourd material, where PI is the parent, P2 is the mother, 1–2 is SL-7 and XDJQ-1, 1–20 is 20 parts of high-SSC wax gourd material, and 21–50 is 30 parts of low-SSC wax gourd material.

**Table 1 ijms-23-06999-t001:** List of genes in the QTL interval.

Gene ID	Whether There Are Nonsynonymous Mutations in Coding Sequences (CDS)	Physical Location	Gene Annotation
Bch02G016850	No	Chr2:58282903-58284486(−)	multidrug and toxin extrusion protein 2-like
Bch02G016860	No	Chr2:58309654-58310180(+)	uncharacterized protein LOC103485927
Bch02G016870	No	Chr2:58334921-58338296(+)	hexokinase-2-like
Bch02G016880	No	Chr2:58338536-58338820(−)	-
Bch02G016890	No	Chr2:58339177-58342311(−)	receptor-like protein kinase 5
Bch02G016900	No	Chr2:58423992-58426566(−)	mitochondrial import inner membrane translocase subunit Tim17-like
Bch02G016910	No	Chr2:58437320-58440270(+)	uncharacterized tRNA/rRNA methyltransferase MAV_0574-like
Bch02G016920	No	Chr2:58443425-58450331(+)	uncharacterized protein LOC103492261
Bch02G016930	No	Chr2:58452533-58454823(+)	uncharacterized protein LOC101220646
Bch02G016940	Yes	Chr2:58474310-58497129(+)	putative ABC transporter B family member 8
Bch02G016950	No	Chr2:58497772-58502600(−)	probable protein phosphatase 2C 6-like
Bch02G016960	Yes	Chr2:58563979-58569234(−)	NADP-dependent malic enzyme
Bch02G016970	No	Chr2:58609828-58644783(−)	probable RNA-dependent RNA polymerase 5 isoform X2

## Data Availability

The data presented in this study are available in this article and as Appendix A.

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
