# Peer review of "Fine Mapping and Functional Analysis of Major Regulatory Genes of Soluble Solids Content in Wax Gourd (*Benincasa hispida*)"

_ijms, 2022, doi:10.3390/ijms23136999_

Round 1

Reviewer 1 Report

General Comments:

If would be of benefit to the readers if the authors would include a bit more in the Introduction about the crop itself. That is, what is the significance of this crop in China, and worldwide? How is it grown. How is it utilized?  Etc.

Line 43-44. Cited reference states that sugar accumulates in the later stages. The results indicate that is not the case – they increase and then decrease.  Explanation?

The authors might consider mentioning that Brix is commonly used as an indicator of SSC.

Line 48 Add a reference after “preference.”

Lines 76 to 87 Is this necessary?

Line 129 Do the authors mean “melon” or gourd? 

Line 141 “The paper is being published” should be removed.  Is there overlap between the data being presented in this MS with the other publication in press?

Line 208  What is meant by “ 50 wax gourd materials”?

Line 219  Reference(s) to support this - “in wax gourd ”

Line 298 How many fruit were used per rep?  Single fruit?

Define difference between biological rep and technical rep?

Figure 1.  Suggest separating part e on Figure 1 and presenting it as a standalone Figure.

No references are cited in the M&M.  Unusual.

Not convinced that so many references are required or such a detailed comparison with Solanum is necessary. What about citing related studies on the subject of SSC in fruit of other cucurbits?  

Italicize scientific names in References

Additional possibly relevant references:

Garcia-Lozano, M., et al. (2020). Transcriptome changes in reciprocal grafts involving watermelon and bottle gourd reveal molecular mechanisms involved in increase of the fruit size, rind toughness and soluble solids. Plant Molec. Biol. 102, 213-223.

doi: 10.1007/s11103-019-00942-7. 

Jiang, B., et al. (2013). De Novo Assembly and Characterization of the Transcriptome, and Development of SSR Markers in Wax Gourd (Benicasa hispida).  PlosOne https://doi.org/10.1371/journal.pone.0071054.

Krishnamoorthy, V. (2020).  Heterosis studies for growth, yield and quality parameters in ridge gourd [Luffa accutangula (Roxb.) L.]. International J. Chem. Studies. Doi: https://doi.org/10.22271/chemi.2020.v8.i3aq.9672.

Song, W., et al. (2022). Comparative Chloroplast Genome Analysis of Wax Gourd (Benincasa hispida) with Three Benincaseae Species, Revealing Evolutionary Dynamic Patterns and Phylogenetic Implications. Genes 13(3), 461; https://doi.org/10.3390/genes13030461.

Xie, D., et al. (2019). The wax gourd genomes offer insights into the genetic diversity and ancestral cucurbit karyotype. Nat. Commun. 10, 5158 (2019). https://doi.org/10.1038/s41467-019-13185-3.

Author Response

Please refer to the attachment for specific modifications.

1.If would be of benefit to the readers if the authors would include a bit more in the Introduction about the crop itself. That is, what is the significance of this crop in China, and worldwide? How is it grown. How is it utilized?  Etc.

Correct:

Insert the following at line 39:

Wax gourd is the main supply of vegetable varieties in summer and autumn in China, which is rich in nutrients and has both medicinal and health benefits[3]. Wax gourd has good storage resistance and long shelf life, so it has become one of the main vegetables that regulate the annual equilibrium supply in the market[4].

References

[3] Song, W.C.; Chen, Z.M.; He, L.;Feng, Q.; Zhang, H.R., Du, G.L., Shi, C., Wang, S.. Comparative Chloroplast Genome Analysis of Wax Gourd (Benincasa hispida) with Three Benincaseae Species, Revealing Evolutionary Dynamic Patterns and Phylogenetic Implications. Genes, 2022,13:461. https://doi.org/10.3390/genes13030461.

[4] Jiang, B.; Liu, W.R.; Peng, Q.W.; He, X.M.; Xie, D.S.. Characterization and chromosomal organization of Ty1-copia retrotransposons in wax gourd.Gene,2014,551:26-32. https://doi.org/10.1016/j.gene.2014.08.014.

2:Line 43-44. Cited reference states that sugar accumulates in the later stages. The results indicate that is not the case – they increase and then decrease.  Explanation?

Correct:

Insert the following at line 222:

Among cucurbitaceae crops, the changes of total sugar content in watermelon[34] and melon[35] during fruit development differed greatly among different materials, and some varieties increased first and then decreased, while others increased all the time. In wax gourd, Huang and Xue[36] observed a trend of sugar content rising first and then decreasing, which was consistent with our research results, while Liu et al.[5] observed that the total sugar content of winter melon continued to increase, which may be related to the different materials we selected and the different ways of sugar accumulation.

References

[34] Yativ, M.; Harary, I.; Wolf, S.. Sucrose accumulation in watermelon fruits: Genetic variation and biochemical analysis. Journal of Plant Physiology, 2010, 167: 589-596. https://doi.org/10.1016/j.jplph.2009.11.009.

[35] Ma, M.J.; Li, P.H.; Li, N.N.; Wang, L.J.. Change of Fruit and Sugar Accumulation during Melon Development. Journal of Henan Agricultural Sciences, 2015,44:82-85. https://doi.org/10.15933/j.cnki.1004-3268.2015.09.020.

[36]Huang, Y.W.; Xue, D.Y.. Study on fruit development of Chieh-qua. Journal of Hunan Agricultural College, 1992,18 Suppl: 807-811. https://doi.org/10.13331/j.cnki.jhau.1992.s4.012

[5] Liu, Z.G.; Wang, P.; Chen, Y. Study on Nutrient Components Change during Chieh-qua Fruit.China vegetables, 2014,8:30-33. https://doi.org/10.3969/j.issn.1000-6346.2014.08.010.

3.The authors might consider mentioning that Brix is commonly used as an indicator of SSC.

Correct:

Line 31-32: Soluble solids content (SSC) is an important quality trait of several melon crops, and Brix is commonly used as an indicator.

4.Line 48 Add a reference after “preference.”

Correct:

Add the following references after “preference.”

[6]Wang, M.; Liu, WR.; He, X.M.; Jiang, B.; Lin, Y.E.; Xie, D.S.; Peng, Q.W.Identification; Evaluation and Utilization of Germplasm Resources of Chieh-qua. Guangdong Agricultural Sciences; 2021, 48 :35-41. https://doi.org/10.16768/j.issn.1004- 874X.2021.05.005.

5.Lines 76 to 87 Is this necessary?

Correct:

Delete Lines 76 to 87.

6.Line 129 Do the authors mean “melon” or gourd? 

Correct:

Change “melon” to “wax gourd”

7.Line 141 “The paper is being published” should be removed.  Is there overlap between the data being presented in this MS with the other publication in press?

Correct:

Removed “The paper is being published”. The data used are only used in this paper, and there is no overlap with other journals.

8.Line 208  What is meant by “ 50 wax gourd materials”?

Correct:

Change “50 wax gourd materials” to “50 wax gourd Inbred lines”

9.Line 219  Reference(s) to support this - “in wax gourd ”

Correct:

Add the following references after “preference[5-6].”

[5]Liu, Z.G.; Wang, P.; Chen, Y. Study on Nutrient Components Change during Chieh-qua Fruit.China vegetables, 2014,8:30-33. https://doi.org/10.3969/j.issn.1000-6346.2014.08.010.

[6]Wang, M.; Liu, WR.; He, X.M.; Jiang, B.; Lin, Y.E.; Xie, D.S.; Peng, Q.W.Identification; Evaluation and Utilization of Germplasm Resources of Chieh-qua. Guangdong Agricultural Sciences; 2021, 48 :35-41. https://doi.org/10.16768/j.issn.1004- 874X.2021.05.005.

10.Line 298 How many fruit were used per rep?  Single fruit? Define difference between biological rep and technical rep?

Correct:

Lines 296 to 300 are modified as follows:

4.2. Measurement of SSC in Wax Gourd Flesh

The date of pollination was recorded for each fruit. The fruits of two parents, F1 and F2, were harvested 20 DAP and pressed into homogenates by using a high-speed homog- eniser. The SSC was measured with a handheld refractometer. Two of the parents and F1 were measured 15 plants respectively, each measuring 3 melons and each melon 3 times. F2 plants were measured in 3 melons per plant and 3 times per melon.

11.Figure 1.  Suggest separating part e on Figure 1 and presenting it as a standalone Figure.

Correct:

Take the suggestion.

12.No references are cited in the M&M.  Unusual.

Correct:

Add references in M&M 4.3. Determination of Glucose, Fructose, and Sucrose[22], 4.4. DNA Extraction[53],and 4.9. RT-qPCR Analysis of Candidate Gene[30].

References

[22] Umer, M.J.; Gao, L.; Gebremeskel, H.; Bin Safdar L.; Yuan, P.L.; Zhao, S.J.; Lu, X.Q.; He N.; Zhu, H.J.; Liu. W.G. Expression pattern of sugars and organic acids regulatory genes during watermelon fruit development. Scientia Horticulturae,2020,265:109102. https://doi.org/10.1016/j.scienta.2019.109102.

[53] An, C.Y.; Xie, D.S.; Peng, Q.W.; He, X.M..Comparison of Genomic DNA Extraction Methods in Chieh-qua (Benincasa hispida). Chinese vegetables, 2011, 24: 1-4. https://doi.org/16861/j.cnki.zggc.2011.05.001.

[30] Cheng, Z.K.;Liu, Z.G.;Xu, Y.C.;Ma, L.L.;Chen, J.Y.;Gou, J.Q.;Su, L.W.;Wu, W.T.;Chen, Y.;Yu, Wen J.;Wang, P.. Fine mapping and identification of the candidate gene BFS for fruit shape in wax gourd (Benincasa hispida). Theoretical and Applied Genetics, 2021, 134: 3983-3995. https://doi.org/10.1007/s00122-021-03942-8.

13.Not convinced that so many references are required or such a detailed comparison with Solanum is necessary. What about citing related studies on the subject of SSC in fruit of other cucurbits?  

Correct:

Modify the lines 236 to 248 as follows:

SSC is a general term for all water-soluble compounds, including soluble sugars, organic acids, soluble proteins, and minerals. Changes in these components affect the SSC, among which, the soluble sugar content is the most important. The changes of sugar metabolism during fruit development can affect the changes of soluble sugars and the proportion of various sugars[38]. The sucrose invertase genes Sucr[39] and Lin5 [40] influence sugar accumulation by affecting invertase activity in tomato. Several enzymes involved in sugar metabolism, such as Beta-glucosidase, bidirectional sugar transporter SWEET10-like play the positive role of watermelon rootstock in sweetness of bottle gourd fruits[41]. In addition to sugar metabolism-related genes that regulate sugar accumulation, sugar accumulation-related genes have been found. Sugar accumulation is negatively affected by vacuolar processing enzyme (VPE) [42] and inhibited by the loss of the synthesis of the plant pigment bilitin (P φ B) in tomato fruits [43]. Putative bel1-like homeodomain protein, which may affect sugar accumulation by influencing the growth processes in melon[12]. These results suggest that sugar accumulation involves complex regulatory networks and is not regulated by simple glucose metabolism.

References

[38]Wei, Q.Z.; Wang, Y.Z.; Qin, X.D.; Zhang, YX..; Zhang, Z.T.; Wang, J.; Li, J.; Lou, Q.F.; Chen, J.F. An SNP-based saturated genetic map and QTL analysis of fruit-related traits in cucumber using specific-length amplified fragment (SLAF) sequencing. BMC Genomics, 2014, 15:1158. https://doi.org/10.1186/1471-2164-15-1158.

[39]Ma, L.L.; Liu, Z.G.; Cheng, Z.K.;,Gou, J.Q.; Chen, J.Y.; Yu, W.J.;Wang, P. Identification and Application of BhAPRR2 Controlling Peel Colour in Wax Gourd (Benincasa hispida). Frontiers in Plant Science,2021,12:716772. https://doi.org/10.3389/fpls.2021.716772.

[40]Cheng, Z.K.;Liu, Z.G.;Xu, Y.C.;Ma, L.L.;Chen, J.Y.;Gou, J.Q.;Su, L.W.;Wu, W.T.;Chen, Y.;Yu, Wen J.;Wang, P.. Fine mapping and identification of the candidate gene BFS for fruit shape in wax gourd (Benincasa hispida). Theoretical and Applied Genetics;2021;134:3983-3995. https://doi.org/10.1007/s00122-021-03942-8.

[41]Jiang, B.;Liu, W.R.; Xie, D.S.; Peng, Q.W.; He, X.M.; Lin, Y.E.; Liang, Z.J. High-density genetic map construction and gene mapping of pericarp color in wax gourd using specific-locus amplified fragment (SLAF) sequencing. BMC Genomics, 2015, 16:1035. https://doi.org/10.1186/s12864-015-2220-y.

[42]Liu, W.R.; Jiang, B.; Peng, Q.W.; He, X.M.; Lin, Y.E.; Wang, M.;Liang, Z.J.; Xie, D.S.; Hu, K.L. Genetic analysis and QTL mapping of fruit-related traits in wax gourd (Benincasa hispida). Euphytica,2018,214:136. https://doi.org/10.1007/s10681-018-2166-7.

[43]Feng, Y.; Li, C.Q.;Zhu, L.Y.;Liu, Y.T.;Yang, X.D.;Zhang, Y.Y.;Zhang, H.;Zhu, W.M.. Research Progress of Soluble Solids Content in Tomato. Molecular Plant Breeding;2021.Available online: https://kns.cnki.net/kcms/detail/46.1068.S.20210122.1747.022.html (accessed on 25 January 2021)

[12] 12.   Cheng, J.T.; Wen, S.Y.; Xiao, S.; Lu, B.Y.; Ma, M.R.;Bie, Z.L..Overexpression of the tonoplast sugar transporter CmTST2 in melon fruit increases sugar accumulation. Journal of Experimental Botany; 2018;69:511-523. https://doi.org/10.1093/jxb/erx440.

14.Italicize scientific names in References

Correct:

Take the suggestion.

Reviewer 2 Report

The manuscript is well-written, only marginal comments:

line 36, 48, 76:  ? Italics

line199, 204: ...figure...

line 523:  I didn´t find this reference (48) in the text... 

Author Response

  1. line 36, 48, 76: ? Italics

Correct: change to italics.

  1. line199, 204: ...figure...

Correct: change ‘figue’ to ’figure’.

  1. line 523:  I didn´t find this reference (48) in the text... 

Correct: the reference numbers have been revised.
